# Using Natural Isotopes for the Environmental Tracking of a Controlled Landfill Site for Non-Hazardous Waste in Liguria, Italy

**DOI:** 10.3390/ijerph22040528

**Published:** 2025-03-31

**Authors:** A. Izzotti, A. Pulliero, Z. Khalid, O. Ferrante, E. Aquilia, S. Sciacca, G. Oliveri Conti, M. Ferrante

**Affiliations:** 1Department of Experimental Medicine, University of Genoa, 16132 Genoa, Italy; 2Department of Health Sciences, University of Genoa, 16132 Genoa, Italy; zumama.khalid@gmail.com (Z.K.); oriana.ferrante1988@gmail.com (O.F.); 3Department of Medical, Surgical and Advanced Technologies “G.F. Ingrassia”, University of Catania, 95125 Catania, Italy; erica.aquilia@unict.it (E.A.); tsciacca42@gmail.com (S.S.); gea.olivericonti@unict.it (G.O.C.); marfer@unict.it (M.F.)

**Keywords:** radioisotopes and stable isotopes, landfill, non-hazardous solid waste, environmental analyses and monitoring, public health, ecosystem quality

## Abstract

The application of natural radioisotope and stable isotope tracing represents a novel, sensitive method for confirming the presence of environmental contamination due to leachate water from solid waste landfills. This study aimed to employ this approach to assess the efficiency of containment measures and the potential environmental impact in the vicinity of a landfill designated for non-hazardous waste disposal. We collected leachate water samples from two distinct areas: one currently active, and another exhausted. In February, May, August, and November 2022, we collected deep water samples from a nearby stream utilizing piezometers, both upstream and downstream from the facility. We examined deuterium and tritium radioisotopes via liquid scintillation, and stable isotope oxygen-18 via ratio mass spectrometry. The results revealed the presence of anthropogenic radioisotopes within the landfill, with higher concentrations in the active site. No radioisotopes or stable isotopes above the natural background were identified in any of the samples obtained from outside. The levels of tritium were found to correlate with rainfall in the samples collected inside, but not in those obtained outside. These findings provide evidence of the effectiveness of the active structural, managerial, and procedural containment measures and the absence of environmental contamination stemming from the studied site, reinforcing the value of the responsible management of non-hazardous waste and its limited impact on the surrounding environment. The reported results highlight the utility of performing radioisotope and stable isotope monitoring not only inside but also outside the landfill, and analyzing the relation via pluviometry.

## 1. Introduction

The landfill known as ‘La Filippa’ [1] stands as a paramount model for the responsible disposal of non-hazardous waste, exemplifying the utilization of the best available technologies to address a multifaceted issue integral to the circular economy: the accumulation of solid waste that surpasses the capacities of reuse, recycling, and recovery [2]. The distinctive approach adopted for waste management at this site embodies a holistic operational paradigm, encompassing design, management, and environmental considerations in a broad context [3].

This innovative approach adopted by ‘La Filippa’ holds substantial scientific significance in assessing its capability to mitigate the adverse impacts on the environment and the well-being of neighboring areas.

Rather than detracting from the environment or health of nearby communities, it strives to enhance their quality. The landfill’s operational globality emphasizes a commitment to sustainable practices, seeking to minimize the ecological footprint associated with waste disposal. The incorporation of advanced technologies and strategic planning in waste management at ‘La Filippa’ underscores the dedication to reducing the environmental consequences of solid waste disposal, ultimately contributing to the overarching goals of the circular economy, leading to positive outcomes for both the environment and society at large [1].

Radioisotope tracing is a cutting-edge environmental monitoring technique that offers valuable insights into the migration and dispersion of leachates and landfill fluids, both within and beyond the boundaries of landfill facilities [4]. This method leverages radioisotopes found in environmental waters, specifically focusing on isotopes of oxygen and hydrogen. For hydrogen, deuterium (^2^H) and tritium (^3^H) radioisotopes are employed, while stable oxygen isotope analysis utilizes oxygen-18 (^18^O) alongside the more common oxygen-16 (^16^O) [4,5]. In natural aquatic systems, all these isotopes are exceptionally scarce. However, within the context of anthropogenic activities and waste disposal, they become relatively more prevalent in leachate waters, particularly in the context of landfill waste and the percolation water it generates. Isotope tracing has emerged as one of the most advanced and sensitive methods for assessing and spatially mapping the movement of leachate water within and beyond landfill sites. Remarkably, it achieves this without any adverse impact on the surrounding environment, relying solely on the collection of environmental water samples [6,7]. This method empowers us to glean critical information about the efficiency of waste neutralization processes within the landfill, as well as whether leachate waters extend into the neighboring territories. For instance, it can pinpoint groundwater pollution or, conversely, the absence of pollution. This is accomplished by monitoring changes in tritium activity relative to its natural baseline levels. By doing so, it becomes possible to estimate the proportion of environmental water mingling with leachate emanating from landfills [8].

Isotope tracing is an invaluable tool not only for environmental monitoring but also for assessing the efficacy of landfill management practices. It aids in ensuring that the integrity of ecosystems and surrounding areas remains uncompromised [9]. This innovative technique exemplifies a commitment to responsible waste disposal and the safeguarding of our natural environment. By offering precise insights into the movement of leachates, it plays a pivotal role in safeguarding water resources and protecting the overall ecological balance [4]. Presently, there exists a notable absence of standardized protocols for utilizing the radio isotopic approach in the environmental monitoring of landfills. In particular, the legislative framework, such as Legislative Decree 36/2003, while stipulating a list of mandatory chemical parameters for monitoring aquifers in proximity to landfill sites, does not encompass the analysis of radioisotope ratios. This omission stands in contrast to the consensus within the international scientific literature, which frequently underscores the efficacy of radioisotopes as unequivocal tracers for detecting leachate contamination in landfills (IAEA/UNESCO, 2000) [10]. At the core of an isotopic investigation into water resource contamination lies the fundamental distinction between the isotopic compositions of landfill wastewater (leachates) and rainwater [11].

Unlike chemical parameters, radioisotope ratios exhibit striking differences between leachates and rainwater. These disparities are so pronounced that even minute levels of contamination can be readily identified. As a result, this method’s sensitivity in discerning the presence or absence of environmental contamination is unparalleled, surpassing the capabilities of traditional chemical analysis techniques [12]. The unique advantage of radioisotope tracing in landfill monitoring lies in its ability to detect the subtlest shifts in isotopic composition, offering a level of precision that is unattainable through conventional chemical analyses, and permitting the fast detection and minimization of a landfill structural issue, while also identifying potential threats to water resources [12,13]. By integrating radioisotope tracing into environmental monitoring protocols, we can enhance our capacity to protect ecosystems and ensure the responsible management of landfills while also aiming to maintain citizens’ health. This approach not only bolsters the scientific basis for landfill management but also underscores a commitment to safeguarding environmental health [4,14].

This study embarked on the utilization of radioisotope tracing both within the confines of and in the areas adjacent to ‘La Filippa’, the definitive disposal site for non-hazardous waste. Its primary objective was to assess the efficiency of the containment strategies implemented and the inherent dynamics of waste neutralization. La Filippa, categorized as a landfill for the disposal of non-hazardous waste with low organic or biodegradable content, operates in accordance with Legislative Decree 36/03 [15]. This environmental investigation began in 2008, with an initial disposal capacity of 450,000 cubic meters, primarily located on the southern side of the landfill site. As this volume became fully utilized, the final stages of the landfill’s life cycle were initiated, including surface coverage, rainwater management, and vegetation restoration. Simultaneously, a new phase of operation commenced in 2015 on an adjacent site with a disposal capacity of 650,000 cubic meters, which is currently under management [16]. The primary focus of this study was to employ environmental radioisotope tracing in the field to assess the efficacy of containment measures and the kinetics of waste inertization at the examined facility. This cutting-edge approach facilitates a comprehensive understanding of how waste containment and neutralization processes unfold over time, enhancing the landfill’s management and environmental impact assessment [17]. By utilizing radioisotope tracing, the study aimed to contribute valuable insights into the sustainable and responsible disposal of non-hazardous waste at ‘La Filippa’ while minimizing any potential ecological impact on the surrounding areas [18].

## 2. Materials and Methods

### 2.1. Study Design and Collection of Environmental Samples

The sampling was performed during the four seasons of the year 2022, with the aim of evaluating the influence of meteorological and seasonal variables on the results obtained. Samplings were performed in the months of February, May, August, and November. For each site and sampling campaign, 2 L of liquid was collected in inert (glass) containers which were sealed immediately after filling. The samples were then taken to the laboratory and analyzed within 4 weeks of arrival. The criteria used for selecting the sampling points were defined with the goal of performing the following comparisons in radioisotope contamination level: (a) inside vs. outside landfill; (b) inside landfill active sites vs. inert sites; and (c) different influence of pluviometry, rain vs. river.

Accordingly, the samples were collected at six different sampling points, both inside and outside, in areas close to the landfill as detailed below:Piezometer 5 (PZ5), located downstream of the landfill in an external area overlooking the southwest side of the settlement;Piezometer 6 (PZ6), located downstream of the landfill in an external area overlooking the southern side of the settlement;Leachate extraction wells 1 and 2 (S1 + S2), located within the settlement in the no-longer-active southern part;Leachate extraction wells 3 and 4 (S3 + S4), located within the settlement in the currently active northern part;Rio Filippa Monte, with water collected directly from the Rio Filippa in its upstream portion on the northwest side of the settlement;Rio Filippa Valley, with water collected directly from the Rio Filippa in its downstream portion on the southeast side of the settlement.The locations of the sampling points in relation to the orographic characteristics of the settlement are shown in Figure 1.

Six different sampling points, both inside and outside, in areas close to the landfill are reported in Figure 1. Rio Monte (PaM) is an upstream portion on the northwest side of the settlement, and Rio Valle (PaV) is a downstream portion on the southeast side of the settlement. PZ5 is located in an external area overlooking the southwest side of the settlement. PZ6 is located in an external area overlooking the southern side of the settlement. S1 and S2 are in the no-longer-active southern part. S3 and S4 are in the currently active northern part. The geographical area monitored was a valley not a plain, and accordingly, pollutant diffusion is not determined by the distance from the source but mainly by the geographical situation. For this reason, sampling points outside the landfill belonging to a stream that brings water downstream to the nearest urban settlement were selected. Sites nearby the landfill were preferred to maximize the sensitivity of the monitoring campaign because the probability of finding contamination was higher than in sites located at a greater distance from the source.

Two radioisotopes (^2^H, ^3^H) and the ^16^O/^18^O ratio were analyzed via alpha/beta liquid scintillation in the collected water samples and isotope ratio mass spectrometry (IRMS), respectively. The analysis of radioisotopes was performed according to ISO 11704 and ISO 17025 standards [19,20]. An aliquot of water sample was acidified with nitric acid, heated to 80 °C under stirring to remove dissolved ^222^Rn, and transferred to a vial with the cocktail of Ultima Gold^TM^ LSC AB scintillation (8:12 volume ratio) (MERCK, Darmstadt, Germany). Liquid scintillation counting was performed using the ultra-sensitive Quantulus 1220 counter (PerkinElmer, Waltham, MA, USA) with calibration parameters and alpha/beta discrimination optimized with ECKERT&ZIEGLER 90Sr and NIST 241Am reference materials. The procedure used for the analysis of each isotope is as follows:**δ ^16^O/^18^O.**

The analysis was performed according to ISO 13166 and ISO 17025 standards [20,21]. An aliquot of water sample was added with Uranium-232 tracer and purified by precipitation with calcium phosphate and Eichrom UTEVA^R^ resin (Eichrom Technologies, Llc., Lisle, IL, USA) before electrolytic deposition on plates of stainless steel. Counting was performed via alpha spectrometry with AMETEK Alpha Duo-Ensemble chambers (ORTEK, Atlanta, GA, USA). The analysis was performed using a Delta V Advantage isotope mass spectrometer coupled to a Flash 2000 HT elemental analyzer via a ConFlo IV (Thermo Fisher Scientific, Waltham, MA, USA). For memory correction, each sample was injected six consecutive times. Normalization to the international reference scale was performed using at least three certified reference materials with different isotopic signatures, each analyzed in triplicate. Quality control was performed by inserting a known–unknown sample at the beginning, middle, and end of each analysis sequence. The results are reported in delta (δ)‰ notation with respect to the international reference scale V-SMOW.


**^2^H**


The analysis was performed according to the ISO 13164-4 standard [22]. An aliquot of water sample was transferred to vials with ProScint Rn scintillation cocktail (10:10 volume ratio). Liquid scintillation counting was performed using the ultrasensitive Quantulus 1220 counter (PerkinElmer, Waltham, MA, USA) with optimized calibration with NIST 226Ra reference material.


**^3^H**


The analysis was performed according to ISO 9698 and ISO 17025 standards [20,23]. Two aliquots of water sample were transferred into two vials with Ultima Gold AB scintillation cocktail (8:12 volume ratio); an aliquot was spiked with a known amount of NIST H3 standard solution to evaluate the counting efficiency with the internal standard method. Liquid scintillation counting was performed using the ultrasensitive Quantulus 1220 counter (PerkinElmer, Waltham, MA, USA).

### 2.2. Measuring Unit

In accordance with the international scientific literature, the results obtained for the ^3^H isotope were expressed as a concentration in terms of Bequerels per liter of sample (Bq/l). These results yielded a positive value (one Bq is 1 disintegration per second). The results obtained for the ^2^H and ^18^O isotopes were expressed with reference to the Vienna Standard Mean Ocean Water (VSMOW) standard. This parameter quantifies the relative presence of the radioisotope compared to the quantity of its stable non-isotopic atomic form present in the sample. The ratio between ^1^H (non-isotopic stable atomic form) and ^2^H (radioisotope) and between ^16^O (non-isotopic stable atomic form) and ^18^O (isotope) is therefore evaluated for each isotope. The VSMOW reference consists of ocean water with ratios of radioisotopic species and non-isotopic stable forms defined as follows for the isotopes analyzed:^2^H/^1^H = 155.76 ± 0.1 ppm (ratio 1/6420);**^16^O/^18^O** = 2005.20 ± 0.43 ppm (ratio 1/499).

The variation in the ratio between the stable non-isotopic atomic form and the radioisotope analyzed in the samples under examination compared to the reference standard constituted by VSMOW was then calculated. The results obtained are expressed as negative numbers, and a higher absolute value therefore corresponds to a smaller quantity and vice versa.

### 2.3. Comparison with Rainfall Data

To evaluate whether precipitation could modify the environmental diffusion of radioisotopes in the areas inside and outside the landfill, we performed a comparison between the radioisotope tracing values and the pluviometric determinations detected in the same period in which the sampling was performed.

### 2.4. Comparison with Waste Delivery Flows

To evaluate whether the quantitative variations in the waste flows delivered to the landfill could modify the environmental diffusion of radioisotopes in the internal and external areas near the landfill, we performed a comparison between the values of the waste deliveries recorded in the same period in which the sampling and isotope tracing results were performed.

## 3. Results

### 3.1. Radioisotope Analysis

The results obtained reported in Table 1 and Table 2 indicate that in the active areas of the settlement (leachate S3 + S4), the value of all three radioisotopes analyzed is significantly higher than that found in the other sampling points (*p* < 0.05 for ^18^O and ^2^H, *p* < 0.001 for ^3^H, as assessed using the nonparametric Kruskal–Wallis test). This result is independent of seasonality and is detectable in all four seasonal harvesting campaigns during the year.

The quantity of anthropogenic isotopes detected in the active landfill volume (leachate S3 + S4) is also significantly greater than that detected in the volume no longer active within the settlement (leachate S1 + S2) (*p* < 0.01 as assessed using nonparametric Mann–Whitney U tests). This result is also not influenced by seasonality as this difference is observable in all four different monitoring campaigns.

However, no difference was detected compared to the background values in the concentrations of isotopes in the samples taken in the areas outside the settlement (PZ5 and PZ6) and in those taken from the Rio both upstream and downstream of the settlement.

### 3.2. Comparison with Rainfall Data

To evaluate whether precipitation could modify the environmental diffusion of radioisotopes in the internal and external areas near the landfill, we performed a comparison between the values of the radioisotope tracing and the pluviometric determinations detected in the same period in which the sampling was performed. In fact, the production of leachate is closely linked to rainfall and not only to the quantity and quality of the waste delivered. Annual rainfall in the site under study normally varies between 800 mm and 1200 mm. Rainfall has an absolute maximum in autumn and a relative maximum in spring. The year 2022 was a particularly dry year at the site, with an overall annual rainfall figure of 411 mm recorded (sum of all 12 monthly rainfall measurements). The quantity of monthly rainfall in the 4 months in which the radioisotopic survey was performed and the quantity of the relative leachate production of the landfill in the active site (S3 + S4) and in the no-longer-active site (S1 + S2) within the settlement are shown in Table 3.

A correlation analysis (Spearman regression analysis r test) was then performed between these rainfall variables, and the quantities of radioisotopes were detected. The correlation between precipitation and quantity of leachate was significant (*p* < 0.05) only in the active site of the landfill (S3 + S4), not in the inactive one (S1 + S2) (Figure 2).

The correlation between precipitation and radioisotope quality was analyzed comparatively at sampling points inside and outside the landfill for the radioisotope ^3^H, the only one with positive values that can be used in the Spearman test. The results obtained are shown in Figure 2.

The results obtained reveal the maximum correlation (*p* < 0.01) between the radioisotope concentration and rainfall in the active site (S3 + S4), while the correlation exists at less robust levels (*p* < 0.05) in the inactive site (S1 + S2), and it is completely missing in both sites external to the settlement (PZ5, PZ6).

The correlation between precipitation and leachate quality was significant only in the active volume of the landfill (S3 + S4), not in the inactive one (S1 + S2). This correlation is totally missing in both piezometers downstream from the settlement (PZ5, PZ6). This result probably derives from the fact that the rains directly hit and wash away the active volume, while the inactive volume is equipped with a suitable final coverage which makes it less easily a direct target of the rainfall fallout. No variation in isotope values was found in points outside the landfill, even in periods of the year with greater rainfall (lack of correlation between ^3^H and rainfall data). This result indicates that the BAT and the structural measures implemented for the drainage of the leachate water produced inside the landfill are effective in containing the contamination inside the landfill without there being negative effects outside.

### 3.3. Comparison with Waste Delivery Flows

To evaluate whether or not the quantitative variations in the waste flows delivered to the landfill were capable of modifying the environmental diffusion of radioisotopes in the internal and external areas close to the landfill, we performed a comparison between the values of the waste deliveries recorded in the same period in which the sampling was performed, and the results of the isotopic tracing were obtained relating to the leachate of the active landfill volume (S3 + S4) and relating to the piezometric waters and those of the Rio (Table 3).

The correlation between contributions and radioisotope tracing for tritium in the different sites sampled both inside and outside the settlement was therefore evaluated. The results obtained are shown in Figure 3.

A significant (*p* < 0.05) correlation between tritium and the quantity of contributions was detected in the active site (S3 + S4). On the contrary, no correlation between tritium and the quantity of contributions was observed in the two piezometers downstream of the settlement (PZ5, PZ6). All the results obtained indicate that the anthropogenic radioisotopes present in the landfill are not released outside the plant.

## 4. Discussion

### 4.1. Results Analysis and Interpretation

The environmental tracking of anthropogenic isotopes within the examined settlement has provided significant insights into the effectiveness of containment measures and the behavior of isotopes in the environment. This study primarily focuses on the presence and distribution of environmental quality indicators inside and outside the plant. It is important to note that the containment measures, including procedural, structural, and management aspects, have been implemented to prevent the release of potentially polluting liquid derivatives, such as leachates, into the surrounding environment. The results show that these containment measures have been successful, as the radioisotopes are primarily detectable within the landfill, with little to no presence outside the plant.

The presence of isotopes within the plant and their absence or natural background values outside the facility suggest that the containment measures have been effective in isolating anthropogenic waste and associated liquid byproducts. This is crucial for preventing environmental contamination and protecting the surrounding ecosystem. It is worth noting that this pattern of radioisotope distribution holds true for both external piezometers near the plant and the waters of the Rio, when sampleable. Furthermore, there is no significant variation in isotope concentrations in the Rio’s waters between sampling points upstream and downstream of the settlement. This finding further confirms that there is no release of anthropogenic materials from the landfill to the areas outside the plant, reaffirming the effectiveness of the containment measures in place.

Among the three monitored isotopes (tritium, deuterium, and oxygen-18), tritium plays a pivotal role due to its half-life of 12.32 years. Tritium monitoring is an effective indicator to assess contamination between leachate and environmental surface and groundwater. Elevated tritium values in groundwater, surpassing typical values found in rainwater, would indicate external contributions, implying that contamination has occurred. In this study, tritium levels were consistently low or undetectable outside the plant, further supporting the efficacy of the containment measures.

Deuterium, on the other hand, is particularly informative about the biological inertization process of landfill waste. Leachate from landfills can be enriched in deuterium because anaerobic processes in these environments lead to the preferential use of hydrogen (^1^H) over deuterium (^2^H) by methanogenic bacteria. This results in higher deuterium concentrations in leachate compared to natural waters. Monitoring deuterium within the landfill reveals valuable insights into the kinetics of waste mineralization on the site.

Comparing the results of radioisotope tracing for deuterium within the active volume of the landfill (S3 + S4) with the inactive site (S1 + S2) provides essential information about the inertization of waste. The noticeable difference in deuterium values between these two sites, with levels approximately two times lower in the inactive site, highlights the dynamic nature of waste inertization. It is important to note that the inactive site was opened in 2008, and the radioisotopic environmental monitoring took place in 2022.

This information, combined with reference background values from external sites, allows us to estimate the rate at which the plant can reduce the content of anthropogenic radioisotopes present in the waste to natural background levels. The dynamics of inertization, calculated based on deuterium data, are presented in Figure 4.

Figure 4 illustrates the quantitative dynamics of waste inertization within the examined plant. It provides insights into how long it takes for the plant to return the content of anthropogenic radioisotopes in the waste to natural background values. This information is invaluable for understanding the long-term environmental impact of the landfill and ensuring that the waste disposal facility remains effective in containing potentially harmful materials.

The results of the environmental tracking of isotopes within the examined settlement demonstrate the successful implementation of containment measures to prevent the release of anthropogenic waste and leachates into the surrounding environment. Tritium monitoring confirms the absence of contamination outside the plant, while deuterium monitoring provides insights into the dynamics of waste inertization. The presented quantitative dynamics of inertization highlight the long-term effectiveness of the landfill in containing waste and minimizing its environmental impact. This study underscores the importance of ongoing monitoring and containment measures in waste disposal facilities to protect the environment and public health.

The estimation of inertization time within the examined landfill provides critical insights into the long-term environmental sustainability of waste management practices. Two kinetic models, linear and logarithmic, have been considered to evaluate the time required for the reduction in anthropogenic isotopes to reach background levels within the active landfill.

In the context of linear kinetics, the model suggests that background values will be achieved after approximately 19 years. This means that over this period, the concentration of anthropogenic isotopes in the landfill will steadily decrease, approaching the natural background levels. This estimation offers a relatively optimistic perspective on the timeline for inertization.

However, adopting a more conservative approach with logarithmic decrease kinetics, as depicted in Figure 4, extends the estimated time for achieving background values to around 33 years. This approach considers the potential for slower decay rates as the concentration of radioisotopes reduces, reflecting a more precautionary estimate of the inertization process.

The relevance of these estimates becomes apparent when considering the phenomenon of a natural reduction in anaerobic fermentation within the landfill. Although this phenomenon is relatively weak, it persists in the landfill. Current estimates suggest that approximately 15 to 20 years of post-management time are necessary for the complete reduction in fermentation to meet expected environmental quality levels.

The alignment of the estimated inertization time, which averages at around 26 years, with the expected time required for natural reduction reinforces the sustainability of the waste management practices within the landfill. These estimates validate the facility’s ability to meet environmental quality standards over the long term and ensure that the needs of the present generation are satisfied without compromising the prospects of future generations.

In summary, the comprehensive assessment of inertization time, considering different kinetic models and the phenomenon of a natural reduction in anaerobic fermentation, affirms the compatibility of the examined waste disposal facility with the principles of environmental sustainability. It underscores the facility’s capacity to protect the environment and maintain its integrity for the benefit of both current and future generations. This alignment with sustainability principles is essential for responsible waste management and the preservation of environmental quality.

### 4.2. Comparison with the Scientific Literature

The results obtained indicate that the system under examination does not release contaminants in the areas external to it. This result was not obvious, given the very high sensitivity of the monitoring method used; in fact, other works that have used this method in other settlements have obtained different results. For example, radioisotope tracing demonstrated the presence of contamination in the case of two landfill sites in Italy. In the first case, it was a landfill located in Sardinia with an underlying geological composition predominantly of sand and alluvial deposits. This geological situation determines a high level of permeability which makes possible the passage of leachate water into the environmental areas surrounding this type of plant. In the second case, it was a landfill located in Umbria, also characterized by the presence of predominantly sandy underlying geological layers (arenaceous, conglomerate sand, alluvial sand, silt) [24]. The considerations already made regarding the high permeability of this type of soil are therefore valid.

For comparison, the landfill examined in this study is characterized by the presence of a geological barrier of compact marl with a strength, at least, of the order of tens of meters, a situation which determines the absence of permeability. This situation explains the absence of an environmental impact detected by isotope tracing in the areas outside the settlement, unlike what happens in other types of landfills that are not as well positioned regarding the site-specific geological and hydrogeological conditions of the settlement site. This low rate can be attributed to only solid special wastes being processed in the landfill, while no organic waste is present.

It should also be underlined that the settlement examined selects the waste delivered, with the consequent presence of a very low rate of biodegradable organic waste. The lower enrichment of deuterium in the leachate detected in the site under examination, compared to what was instead detected by the study of other authors in the landfills in Sardinia and Umbria [19], is therefore also probably an indication of lower methanogenic activity attributable to the lower presence of biodegradable waste. In fact, a strong presence of deuterium is detectable in the case of landfills with highly biodegradable contributions, as demonstrated in an analysis of a plant in Minnesota, USA [25].

In another study, the contamination of the areas surrounding the Antananarivo landfill, Madagascar, was detected by analyzing the environmental diffusion of tritium. In that case, the underlying geological layers comprised clay, peat, kaolin sand, and sand with increasing grain size. However, the surface clay layer was very thin compared to that present in the landfill examined in our study. In fact, the Madagascar landfill was only 1 m thick compared to a thickness of >50 m found in the La Filippa landfill. In addition to leachate, the authors of the Madagascar landfill study also identified evaporation and condensation fallout as a possible mechanism for the environmental contamination detected in areas outside the landfill. The situation in Madagascar relates to the lack of landfill coverage, the high environmental temperatures, and above all the indiscriminate delivery of waste with a high biodegradable component [26]. Fermentation processes occurring in biodegradable waste may influence the radioisotopic composition of landfill biogas and may be used to verify the progress of biological stabilization of waste [27]. Table 4 summarizes the results and data of the landfill monitoring campaigns reported as compared to those presented in this study (Table 4).

## 5. Conclusions

The findings from our environmental isotope tracing study offer compelling evidence that the La Filippa landfill is an environmentally responsible facility. It has been demonstrated that this landfill effectively safeguards against the contamination of environmental matrices, including surface water, groundwater, and soil. Moreover, the facility proves to be highly efficient in segregating non-hazardous waste from the surrounding environment, ensuring that no detrimental impact is observed outside its confines. The marked reduction in traced radioisotopes within the landfill serves as a clear indicator of the waste’s inertization.

In summary, the La Filippa production site, owing to a combination of robust structural design and effective waste management practices, leaves no discernible footprint on the surrounding areas. Using the highly sensitive environmental radioisotope tracing method underscores the facility’s commitment to environmental stewardship and its dedication to maintaining the quality of the ecosystem beyond its boundaries. These results signify a responsible and sustainable approach to waste management that ensures the needs of the present generation are met without compromising the well-being of future generations. Our results provide evidence of the importance of radioisotope monitoring in evaluating the environmental impact of landfills. The La Filippa landfill utilizes many of the best available intervention technologies to mitigate environmental impact, including impermeabilization, draining geonet, infra-structures for leachate collection, purification, and outflow, as summarized in Figure 5.

However, the efficacy of these interventions should be monitored using sensitive and specific tools. Chemical analyses of environmental matrixes are necessary but are not sufficient and specific enough to provide robust scientific results establishing the environmental impact. Indeed, chemical biomarkers such as nitrite, sulfur, and hydrogen may be influenced by natural or artificial sources outside the landfill. Conversely, the monitoring of natural isotopes represents a highly sensitive and specific tracer of pollution deriving from landfills. Our study demonstrates the importance of performing this monitoring not only inside but also outside the landfill, and analyzing the correlation of isotopes as compared with pluviometry and working status. The correlation between landfill isotopes and pluviometry is of remarkable importance currently because of the frequency of extreme climate events.

Based on the results of this study, the following standardized protocol for using radioisotopes in landfill monitoring is proposed: (a) test both oxygen isotopes and hydrogen radioisotope; (b) collect samples both inside and outside the landfill; (c) compare active and nonactive sites inside the landfill to evaluate the inertization time; (d) compare results with pluviometry.

Radioisotope environmental monitoring represents a new tool for evaluating landfill safety, and offers a specific approach not influenced by environmental backgrounds or other pollution sources as typically happens in the chemical analysis of environmental matrices. Furthermore, it is relatively simple and low-cost, thus representing an efficient tool for maintaining public health. It facilitates not only the evaluation of landfill safety but also the efficacy of corrective measures in the case of unsafe landfills.

Future research directions will be the parallel analyses of isotopes with other pollutants emitted by landfills with reference not only to liquid pollutants but also to gaseous pollutants such as methane.

## Figures and Tables

**Figure 1 ijerph-22-00528-f001:**
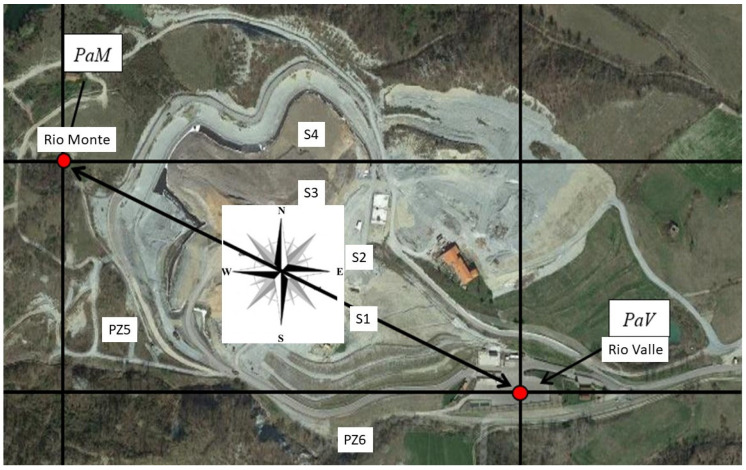
Plan and top view of the monitored site. S1–S4—sampling sites inside the landfill; PZ5–6—sampling sites outside the landfill. PaM—upstream area above the landfill (Rio Monte); PaV—down-stream area below the landfill (Rio Valle).

**Figure 2 ijerph-22-00528-f002:**
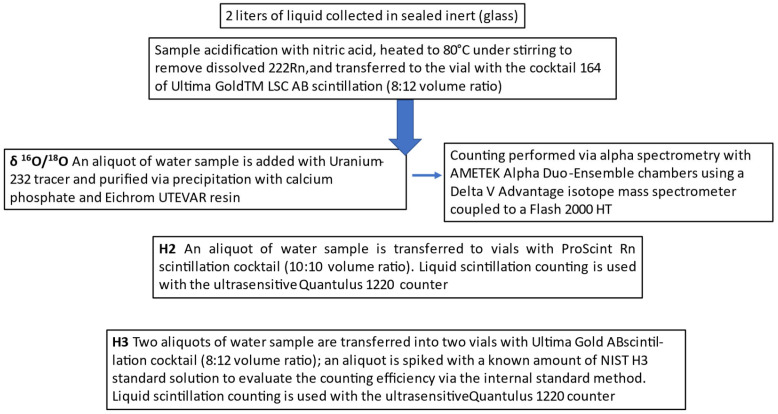
Flow chart.

**Figure 3 ijerph-22-00528-f003:**
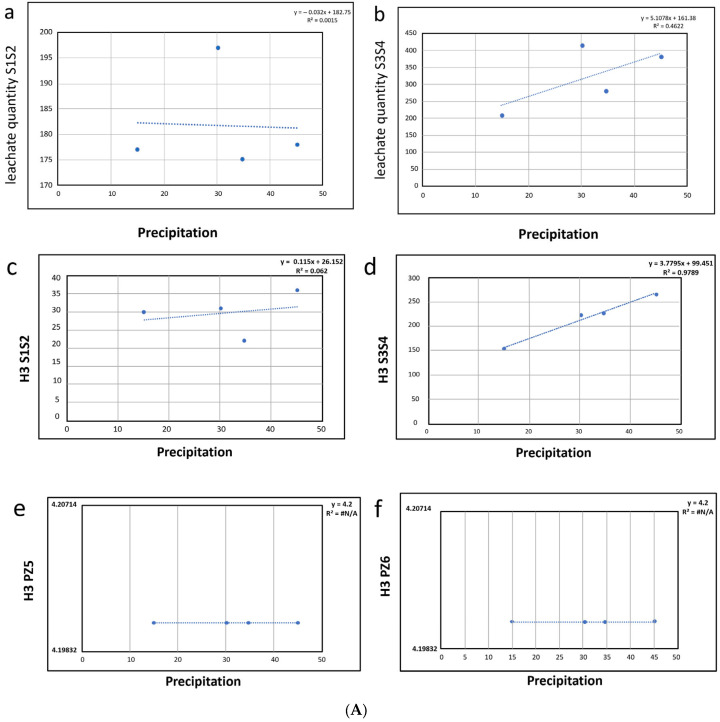
(**A**) Linear regression analyses of the correlation between rain precipitation (x axes) and radioisotope amounts (y axes) in sampling points inside the landfill, either inert (**a**,**c**) or still in use (**b**,**d**); sampling points outside the landfill (**e**,**f**). Significant correlations between rain precipitation and radioisotope amounts in leachate were detected only in active sites inside the landfill. (**B**) Linear regression analyses of the correlation between waste contribution (x axes) and radioisotope (y axes) in sampling points in active sites inside the landfill (**a**,**b**); sampling points outside the landfill (**c**,**d**). Significant correlations were detected only in active sites inside the landfill.

**Figure 4 ijerph-22-00528-f004:**
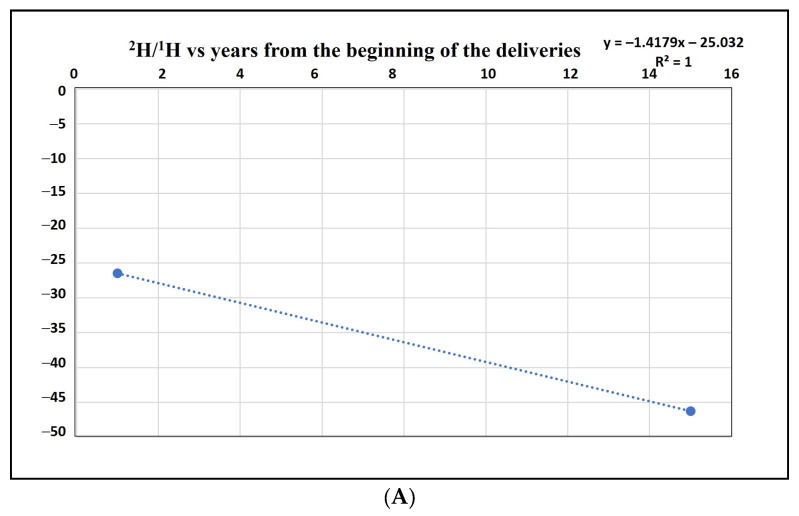
Dynamics of waste inertization within the examined plant as inferred from radioisotope analysis. X axis reports the years of decay; Y axis represents the radioisotope amount (H2/H1 ratio). Both linear (**A. upper panel**) and logarithmic (**B. lower panel**) decrease kinetics were analyzed.

**Figure 5 ijerph-22-00528-f005:**
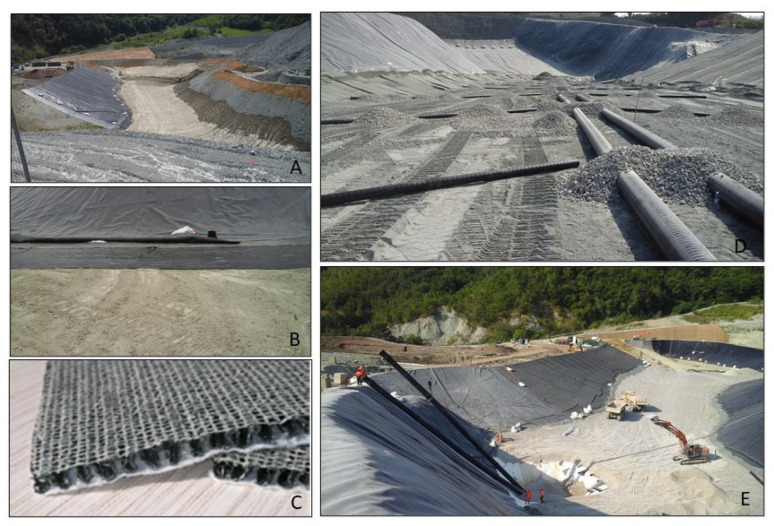
Best available intervention technologies utilized at the La Filippa landfill to mitigate environmental impact. (**A**,**B**) impermeabilization; (**C**) draining geonet, (**D**,**E**) infrastructures for leachate collection, purification, and outflow.

**Table 1 ijerph-22-00528-t001:** Results presented for the February and May collection campaigns, across all sampling points and each of the three monitored radioisotopes.

ID	Bq/l	% VSMOW **	% VSMOW **	Sampling Period
	^3^H—TRIZIO ^§^	δ^18^O/^16^O *	^2^H/^1^H *	February: First Sampling
PZ5	<4.2	−8.15 ± 0.06	−53.3 ± 0.4	
PZ6	<4.2	−8.04 ± 0.02	−53.8 ± 0.02	
S1 + S2	**30 ± 5.0**	−8.11 ± 0.10	−52.2 ± 0.2	
S3 + S4	**152 ± 7.0**	**−6.57 ± 0.06**	**−30.7 + 0.4**	
Rio Monte	<4.8	−8.02 ± 0.04	−52.2 ± 0.2	
Rio Valle	<4.8	−7.99 ± 0.08	−52.3 ± 0.6	
PZ5	<4.2	−7.33 ± 0.04	−54.8 ± 1.8	May: Second Sampling
PZ6	<4.2	−7.54 ± 0.75	−54.5 ± 1.7	
S1 + S2	**30.6 ± 6.1**	−7.03 ± 0.33	−50.1 ± 1.5	
S3 + S4	**224 ± 3.7**	**−5.61 ± 4.32**	**−27.6 ± 0.8**	
Rio Valle	<4.8	−7.78 ± 1.66	−52.9 ± 1.7	
Rio Monte	NP	NP	NP	

^§^ Minimum level of detection 4.2 Bq/l; * (%); ** % Vienna Standard Mean Ocean Water; NP: samples were not received due to the bottle breaking during shipment to the laboratory. Values higher than the background are highlighted in bold character.

**Table 2 ijerph-22-00528-t002:** Results presented for the August and November collection campaigns, encompassing every sampling point and the three monitored radioisotopes.

ID	Bq/l	% VSMOW **	% VSMOW **	Sampling Period
	^3^H—TRIZIO ^§^	δ^18^O/^16^O *	^2^H/^1^H *	August: Third Sampling
PZ5	<4.2	−7.73 ± 0.41	−55.8 ± 3.5	
PZ6	<4.2	−8.86 ± 0.48	−50.5 ± 1.7	
S1 + S2	**22.3 ± 5.10**	−7.01 ± 0.51	−42.9 ± 4.5	
S3 + S4	**228 ± 31.0**	**−5.52 ± 1.41**	**−23.7 ± 2.0**	
Rio Monte	NP	NP	NP	
Rio Valle	NP	NP	NP	
PZ5	<4.2	−7.39 ± 0.33	−50.6 ± 0.50	November: Fourth Sampling
PZ6	<4.2	−7.18 ± 0.51	−50.2 ± 1.00	
S1 + S2	**36.4 ± 7.10**	−6.89 ± 0.13	−40.1 ± 0.40	
S3 + S4	**267 ± 35.0**	**−6.28 ± 0.41**	**−23.8 ± 0.30**	
Rio Monte	NP	NP	NP	
Rio Valle	NP	NP	NP	

^§^ Minimum level of detection 4.2 Bq/l; * (%); ** % Vienna Standard Mean Ocean Water; NP: samples were not received as they were not taken due to the Rio dry period. Values higher than the background are highlighted in bold character.

**Table 3 ijerph-22-00528-t003:** Monthly rainfall and leachate production comparison in active and inactive landfill sites. Monthly contributions of radioisotope detection over four months.

Month	Monthly Rain (mm)	Total Leachate Extracted S1 + S2 (MC) Phase 1 Volume	Total Leachate Extracted S3 + S4 (MC) Phase 2 Volume	Delivery Quantity (Tons)
February	15.00	177	209	8093
May	30.20	197	415	11,925
August	34.80	175	280	4298
November	45.20	178	381	14,445

**Table 4 ijerph-22-00528-t004:** Results and data of the landfill monitoring campaigns.

Site	Geological Structure	Detection Methods	Maximum Distance of Radioisotope Detection Outside the Landfill	Reference
Sardinia island, Italy	Sand and alluvial deposits	H2, O18	Not analyzed	[24]
Umbria region, Central Italy	Arenaceous, conglomerate sand, alluvial sand, silt	H2, O18	Not analyzed	[24]
Antananarivo, Madagascar	Clay, peat, kaolin sand, 1 m thick	H3	700 m	[26]
Ligurian region, North West Italy	Compact marl, >50 m thick	H3, O18 detection	0 m	This study

## Data Availability

All data generated or analyzed during this study are included in this published article. If other researchers wish to replicate this study, they can contact the Corresponding Author directly.

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
