# Peer review of "Using Natural Isotopes for the Environmental Tracking of a Controlled Landfill Site for Non-Hazardous Waste in Liguria, Italy"

_ijerph, 2025, doi:10.3390/ijerph22040528_

Round 1
Reviewer 1 Report
Comments and Suggestions for Authors
To assess the effectiveness of containment measures and the potential environmental impact in the vicinity of a landfill, the natural radioisotope tracing method is applied. The findings provide evidence to enhance the responsible management of non-hazardous waste and its limited impact on the surrounding environment.
The following problems should be addressed in the new version.
1. In Section 2, a flowchart should be included to provide a detailed overview of the used method.
2 The materials discussed in Section 2 should be presented in a table, and the parameters of the testing equipment should be summarized in a table.
3 Figures 2 and 3, are unclear and need to be replotted.
4. Why is there only Subsection 4.1 in Section 4?
5. In Section 4, comparisons between different methods should be presented in a table.
6. In the Conclusion section, what is the contributions of this article and the future research directions should be given out.
Author Response
Dear Editors,
We would like to thank you for considering the manuscript entitled “Environmental tracking using natural radioisotopes of a con-trolled landfill site for non-hazardous waste in Liguria, Italy” by Izzotti A. et al., and for sharing the Reviewers’ comments that certainly helped in improving the quality of the manuscript (ID: ijerph-3390135). We appreciated the Reviewers’ comments, and we revised the manuscript accordingly. Please find enclosed to the submission of the revised version of the manuscript the point-by point reply to the Reviewers’ comments. For clarity’s sake, changes in the revised MS are wrote in yellow color.
We hope that this revised version of our MS will be now suitable for publication in the ijerph.
Accordingly, we prepared a revised version of the manuscript acknowledging Referees’ and Editor’s comments as below specified:
Reviewer 1:
To assess the effectiveness of containment measures and the potential environmental impact in the vicinity of a landfill, the natural radioisotope tracing method is applied. The findings provide evidence to enhance the responsible management of non-hazardous waste and its limited impact on the surrounding environment. The following problems should be addressed in the new version.
Comment 1: In Section 2, a flowchart should be included to provide a detailed overview of the used method.
ANSWER 1. We thank the Reviewer for the suggestion. A flowchart has been included in Section 2.
Comment 2: The materials discussed in Section 2 should be presented in a table, and the parameters of the testing equipment should be summarized in a table.
ANSWER 2. We agree with this comment. A table has been added as suggested.
Comment 3: Figures 2 and 3, are unclear and need to be replotted.
ANSWER 3. Figure 2 has been removed and Figure 3 has been replotted as requested .
Comment 4: Why is there only Subsection 4.1 in Section 4?
ANSWER 4. Subsection 4.1 and 4.2 have been added as suggested.
Comment 5: In Section 4, comparisons between different methods should be presented in a table.
ANSWER 5. A comparative Table reporting the results and data of the different landfill monitoring methods reported in Section 4 has been added (Table 4).
Comment 6: In the Conclusion section, what is the contributions of this article and the future research directions should be given out.
ANSWER 6. A paragraph and a Figure (Figure 4) highlighting the scientific contributions of this article and the future research directions has been added as suggested both in Conclusion and in the Abstract.

Reviewer 2 Report
Comments and Suggestions for Authors
Overall this work can be considered a case study and the application of radioisotope tracing can be somehow novel. However, the paper is not properly edited and the quality of figures is really low. Also, the writing needs to be improved in many areas.
1- In technical paper, the use of “we” (line 16 , line 18) should be avoided. All writing should be conducted in the passive form.
2- Section 2: How the six sampling points were selected?
3- Why the authors did not choose control sites that are far from the landfill. This way, a broader baseline can be established.
4- Line 143. A sentence cant start with “6”. It should be stated “six”. Any numbers higher than 10 can be mentioned as is.
5- Figure 1. Caption is missing. General comment. many figures are missing captions, and in other places, captions were placed far from the figures. Also some figure in page 7, 9 are not legible at all.
6- Line 434 – 440 The authors are mentioning a low rate of biodegradable waste. How were these rates measured?, and how do this rate of biodegradable waste compare to other landfill sites?
7- The authors can improve the quality of their work by proposing standardized protocols for using radioisotopes in landfill monitoring.
8- The conclusion of this work is that there is absence of detectable contamination, so is there any suggestions for any change regarding local public health policies?
9- Figure 4 and 5. The plots contain only 2 points. So it is trivial to have R2 =1. Also in these figures, the x and y axis titles are missing. One more thing, why the authors are using a linear equation in fig 4 and logarithmic in figure 5?
Comments on the Quality of English Language
The paper is not properly edited.
Author Response
Reviewer 2
Overall this work can be considered a case study and the application of radioisotope tracing can be somehow novel. However, the paper is not properly edited and the quality of figures is really low. Also, the writing needs to be improved in many areas.
Comment 1: In technical paper, the use of “we” (line 16, line 18) should be avoided. All writing should be conducted in the passive form.
ANSWER 1. Thank you for pointing this out. The lines have been corrected.
Comment 2. Section 2: How the six sampling points were selected?
ANSWER 2. A paragraph has been added reporting selection criteria used for selecting the 6 sampling points
Comment 3. Why the authors did not choose control sites that are far from the landfill. This way, a broader baseline can be established.
ANSWER 3. A paragraph providing details dealing the selection of control sites outside the landfill has been added in Section 2 as suggested. (lines 155-161).
Comment 4. Line 143. A sentence can’t start with “6”. It should be stated “six”. Any numbers higher than 10 can be mentioned as is.
ANSWER 4. Thank you for pointing this out. The sentence has been corrected.
Comment 5. Figure 1. Caption is missing. General comment. many figures are missing captions, and in other places, captions were placed far from the figures. Also some figure in page 7, 9 are not legible at all.
ANSWER 5. Figure 1 caption has been added as suggested. Figure 3 has been reformatted. Figure 3 and 4 caption have been added.
Comment 6. Line 434 – 440 The authors are mentioning a low rate of biodegradable waste. How were these rates measured?, and how do this rate of biodegradable waste compare to other landfill sites?
ANSWER 6. It is now reported that ‘This low rate is since only solid special wastes are processed in the landfill while no organic waste is present’.
Comment 7. The authors can improve the quality of their work by proposing standardized protocols for using radioisotopes in landfill monitoring.
ANSWER 7. A paragraph proposing a standardized protocol for using radioisotopes in landfill monitoring has been added (Conclusion, lines 557-561).
Comment 8. The conclusion of this work is that there is absence of detectable contamination, so is there any suggestions for any change regarding local public health policies?
ANSWER 8. A sentence suggesting the use of radioisotope monitoring to evaluate landfill safety has been added in Conclusion (lines 562-567).
Comment 9. Figure 4 and 5. The plots contain only 2 points. So it is trivial to have R2 =1. Also in these figures, the x and y axis titles are missing. One more thing, why the authors are using a linear equation in fig 4 and logarithmic in figure 5?
ANSWER 9. R2=1 has been deleted from Figures. In the legends it is now reported that upper panel reports a linear equation and lower panel logarithmic equation.
Comment 10. Comments on the Quality of English Language The paper is not properly edited.
ANSWER 10. Professional English revision has been requested to MDPI services.

Reviewer 3 Report
Comments and Suggestions for Authors Dear authors, while reading your text I came to the conclusion that you do not distinguish between stable and radioactive isotopes of oxygen and hydrogen. Ignorance of the basics of the topic of your work has led to a number of inaccuracies in the work, which include the use of incorrect methods, incorrect expression of the results and thus inappropriate interpretation. I suggest that your manuscript to be rejected.Author Response
Reviewer 3:
Comment 1: Dear authors, while reading your text I came to the conclusion that you do not distinguish between stable and radioactive isotopes of oxygen and hydrogen. Ignorance of the basics of the topic of your work has led to a number of inaccuracies in the work, which include the use of incorrect methods, incorrect expression of the results and thus inappropriate interpretation.
I suggest that your manuscript to be rejected.
ANSWER 1.
Dear reviewer, thank you for your valuable comments. Your comments were useful because highlighted a refuse text in the submitted paper concerning the stable isotopes O16 and O18 methodology, in fact , the wrong methodology was revised and corrected in the Revised version of our paper. We analyzed the O16/O18 using the isotope ratio mass spectrometry (IRMS) that it is the reference methodology for stable isotopes detection. We apologize for the error that made not realist the results in the original paper.
The correct text about the O18 analysis was modified in the revised format of the paper. We used, in fact, only referenced methodologies respectively for radioisotopes and stable isotopes.
Also, we have corrected all indications about isotopes and radioisotopes to avoid inappropriate interpretations.

Round 2
Reviewer 1 Report
Comments and Suggestions for Authors
It's OK.
Reviewer 2 Report
Comments and Suggestions for Authors
the authors properly addressed my concerns. i have no reservations
Comments on the Quality of English LanguageEnglish needs improvement in some areas